

# Synoptic and microphysical lifetime constraints for contrails

Sina Maria Hofer[1] and Klaus Martin Gierens[1]

[1]Deutsches Zentrum für Luft- und Raumfahrt, Institut für Physik der Atmosphäre, Oberpfaffenhofen, Germany

**Correspondence:** Sina Maria Hofer (sina.hofer@dlr.de)

**Abstract.**

Contrail lifetime is constrained mainly by the sedimentation of ice crystals into lower levels that are subsaturated, the blowing out of the ice crystals from the parent ice supersaturated regions (ISSRs) by the (horizontal) wind, and the reduction of supersaturation down to subsaturation due to large-scale subsidence. The first of these processes can be characterised by a sedimentation time-scale. The second and third processes can together be characterised by a synoptic time-scale. The synoptic time-scale is determined in this paper with trajectory calculations for air parcels that initially reside in ice supersaturated regions and which either leave these with the wind or where the ice supersaturation itself vanishes. It is crucial to know the time-scales of contrails because their individual effect on climate depends on their lifetime. The distinction between the two time-scales is particularly important for planning flights with alternative fuels for the purpose of the mitigation of contrail effects. This works in particular if sedimentation is the predominant contrail termination process, that is, if the sedimentation time-scale is shorter than the synoptic one. Here we show that both time-scales are of the order of a few hours. Actually, in nature, the three mentioned processes act simultaneously. The combined time-scale is half of the harmonic mean of the two time-scales in separation. Furthermore, we found as a side result that ISSRs only emerge in areas where the normalised geopotential height, $Z^*$, is mainly at least 0.98. For contrail-avoiding flight planning this means that contrail avoidance in regions with $Z^* < 0.98$ is mainly not necessary.

## 1 Introduction

The individual radiative effect of a contrail (instantaneous radiative forcing or energy forcing) is the radiative flux change (infrared and solar radiation) which it causes during its complete lifetime. The lifetime of a single contrail is thus an important characteristic. Contrail lifetimes vary widely; most contrails are short and terminate already after a few minutes. These are contrails that have been formed in air with subsaturated water vapour, that is, where the relative humidity with respect to ice is below 100%. Ice crystals in such an environment sublimate quickly. The contrail lifetime can be as short as a few seconds in very dry air up to a few minutes in slightly subsaturated air (Sussmann and Gierens, 2001). Such contrails are generally considered not climate-relevant. Contrails that are formed within ice supersaturated regions (ISSRs) become older than a few minutes. In fact, they can reach lifetimes exceeding 10 hrs (eg. Minnis et al., 1998; Haywood et al., 2009). These contrails are persistent. They can spread and extend into contrail-cirrus. They are relevant for climate due to their interaction with radiation (Schumann et al., 2012; Wolf et al., 2023) and with nearby natural cirrus clouds (Verma and Burkhardt, 2022) and





due to their effect on the upper-tropospheric water budget (Schumann et al., 2015). All these effects should get stronger with increasing contrail lifetime and thus it is important to consider the lifetime statistics from different viewpoints. Gierens and Vázquez-Navarro (2018) used contrail tracking data from a geostationary satellite to derive lifetime statistics. Taking into

account unseen periods of contrail lifetimes, they derived a mean lifetime of the order of three hours and concluded that about 5% of contrails have lifetimes exceeding 10 hours.

Contrail lifetimes are mainly constrained by three different processes, one of which is known as the microphysical pathway, while the two others form the synoptic pathway (Bier et al., 2017). In the microphysical pathway, contrails are dissolved by sedimentation of their ice crystals into lower drier layers. The synoptic pathway implies that either the air itself which

contains the contrail becomes drier and sub-saturated or that the contrail is blown out of the ISSRs with the wind. Currently, it is unknown which of these pathways dominates or if they occur with similar frequency. This question, however, has some bearing on the use of alternative fuels in a given situation. Alternative fuels generally lead to reduced soot emission (Moore et al., 2017; Voigt et al., 2021), which in turn leads to less but larger ice crystals which sediment faster. This is beneficial for the contrail climate impact, but only effective in the microphysical pathway; in the synoptic pathway the ice crystals sublimate

anyway in the sub-saturated air. The application of (still expensive and not available in large amounts) alternative fuels for the benefit of climate should thus be planned carefully and it would help to know in advance which contrail termination pathway is more likely in the current or forecast weather situation.

In this paper we consider the contrail termination pathways from the viewpoint of two time-scales, the sedimentation time-scale $\tau_{\mathrm{sed}}$ and the synoptic time-scale $\tau_{\mathrm{syn}}$. It will turn out that both time-scales are of the same order of magnitude (a couple

of hours), which is probably the reason why it is not yet known which pathway dominates. Perhaps, it could also imply that in many cases neither of the two pathways dominates. However, if in a given situation the aviation weather forecast would result in $\tau_{\mathrm{sed}} < \tau_{\mathrm{syn}}$, then the microphysical pathway is likely and alternative fuels can effectively be used. On the contrary, if $\tau_{\mathrm{sed}} > \tau_{\mathrm{syn}}$ is predicted, the synoptic pathway will probably dominate, rendering use of alternative fuels inefficient.

The paper is structured as follows: The data used in the study are described in section 2. In section 3 the methods and results

of the time-scale for sedimentation of ice crystals in cirrus and contrails (in section 3.1) and contrail movements out of ice supersaturated regions (in section 3.2) are shown. The results are discussed in section 4. At the end, we conclude in section 5.

## 2  Data

Four times per day the German Weather Service (DWD) provides hourly aviation weather forecasts (WAWFOR data), based on the weather forecast model ICON (Zängl et al., 2015). For the present analysis, the temperature and the relative humidity

with respect to water ($RH_w$) are used to calculate the relative humidity with respect to ice ($RH_i$). In addition, wind data are used, which are given in their zonal and meridional components. Usually, the WAWFOR data inform aviation users about the temperature, humidity, winds, etc. (WAWFOR Package 1). For the German D-KULT project (Demonstrator Klima- und Umweltfreundlicher Lufttransport; Demonstration of climate- and environmentally friendly air transport), additional datasets are produced. They provide information about the potential to form persistent contrails. There is a binary field called the po-



tential of persistent contrails $PPC$ (either 0 or 1). $PPC$ is obtained from the Schmidt-Appleman criterion (SAC) (Schumann, 1996) applying an overall propulsion efficiency of $\eta = 0.365$ and using temperature and relative humidity from the regular forecast. For the compensation of a low humidity bias in the forecast (Gierens et al., 2022), situations with $RH_i > 93\%$ are considered ice (super)-saturated (Hofer et al., 2024). Thus, $PPC = 1$ marks grid points where persistent contrails are possible, and $PPC = 0$ marks all the rest where either no contrails at all or merely short contrails can be formed. The data are available globally and with higher spatial resolution for the European region (EU nest, $0.0625° \times 0.0625°$, approx. $6.5\,\text{km} \times 6.5\,\text{km}$). The latter are used here in an area from $23.5°\text{W}$ to $62.5°\text{E}$ and $29.5°\text{N}$ to $70.5°\text{N}$. We consider the situations at $250\,\text{hPa}$ for three days with different weather conditions: 18/04/2024, 01/05/2024, and 24/05/2024.

The data for the description of the synoptic situations have been obtained from the Pamore system (DWD, 2024) of DWD. They are based on ICON forecasts as well.

## 3   Methods and results

### 3.1   Time-scale for sedimentation of ice crystals in cirrus and contrails

An analytical expression for the time-scale of ice crystal sedimentation from contrails or cirrus clouds can be derived using equations provided by Spichtinger and Gierens (2009, in the following abbreviated SG09).

Let us consider a contrail filling a volume $H \times W \times L$ (height, width, length). Let the ice mass inside this volume be $M = HWL \cdot \varrho q_i$ (with air density $\varrho$ and ice mass mixing ratio $q_i$). The change of $M$ due to sedimentation is given by the sedimentation flux $F_m \times WL$:

$$\left(\frac{dM}{dt}\right)_{\text{sed}} = HWL\varrho\frac{dq_i}{dt} = WLF_m,$$

$$\text{thus} \qquad H\varrho\frac{dq_i}{dt} = F_m. \tag{1}$$

Now, the flux density $F_m$ is $\varrho q_i v_m$ where $v_m$ is the mass-weighted fall speed of an ensemble of ice crystals. The latter is computed in SG09 via general moments $\mu_k$ of the ice crystal mass distribution:

$$v_m = \gamma(m)\mu_{\delta(m)+1}/\mu_1. \tag{2}$$

$\gamma(m), \delta(m)$ are mass-dependent, but piecewise constant. In the range relevant for contrails their values are $\gamma = 63292.4, \delta = 0.57$ (please note that the given value of $\gamma$ is only valid for SI-units, kg, m, s, see table 2 in SG09). Further, we note that $q_i = \mu_1 = N_i \cdot \overline{m}$, with the crystal number per kg of air $N_i$ and the mean ice crystal mass $\overline{m}$. The other moment required is given as $\mu_{\delta+1} = N_i \cdot \overline{m}^{(\delta+1)} r_0^{(\delta+1)\delta/2}$. Here, $r_0$ is a parameter that determines the width of the (lognormal) mass distribution (typically $2 \leq r_0 \leq 4$). Thus, $\mu_{\delta(m)+1}/\mu_1 = \overline{m}^{0.57} r_0^{0.45}$.

Finally, we determine the sedimentation time-scale as $\tau_{\text{sed}}^{-1} = (1/q_i)dq_i/dt$ and find by combining the expressions just derived:

$$\tau_{\text{sed}} = \frac{H}{\gamma \overline{m}^{0.57} r_0^{0.45}}. \tag{3}$$





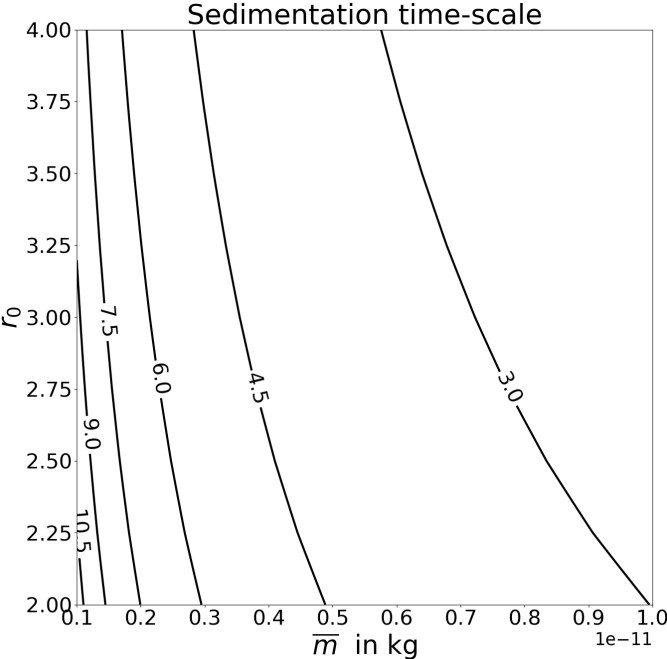

**Figure 1.** Time-scale for sedimentation of ice crystals out of contrails with height $500\,\text{m}$ in hours as function of mean crystal mass $\overline{m}$ and mass distribution width parameter $r_0$. As the time-scale is proportional to contrail height, time-scales for different contrail heights can easily be derived from these curves; for instance, a time-scale of $5\,\text{h}$ for a $500\,\text{m}$ deep contrail would reduce to $3\,\text{h}$ for a $300\,\text{m}$ deep one.

Note, that $\tau_{\text{sed}}$ is in seconds and $\overline{m}$ must be taken in kg (SI-units). For $H = 500\,\text{m}$ and a mean mass of the order $10^{-12}\,\text{kg}$ the sedimentation time-scale is a few hours (say $2.5$ to ten hours).

## 3.2   Contrail movements relative to ice supersaturated regions

### 3.2.1   Time-scale for contrails to leave an ISSR with the wind in theory

The results below will show that the time, $T$, an air parcel resides within an ISSR is Weibull-distributed, that is

$$S(T) = 1 - F(T) = \exp[-(T/T_0)^k].\tag{4}$$

Here, $S(T)$ is the survival function, and $F(T)$ is the cumulative distribution function of $T$ (see Gierens and Vázquez-Navarro, 2018). For the case of sedimentation, we defined a timescale as the time when a fraction of about $e^{-1}$ of the ice mass is left in the contrail. To be consistent with that definition we define the time-scale, $\tau_{\text{syn}}$ for leaving an ISSR as that time, where the survival function reaches the same value, $S(\tau_{\text{syn}}) = e^{-1}$. This time is given by the parameter $T_0$, independently of the exponent

$k$ of the Weibull distribution. Our result is thus:

$$\tau_{\text{syn}} = T_0.\tag{5}$$





The Weibull fits in the paper yield straight lines of the form

$$\ln\ln[1/S(T)] = \beta + k\ln(T/T_u) \tag{6}$$

where $\beta$ is the intercept and $k$ is the slope. $T_u$ is a unit time, e.g. 1 s. Solving for the survival function and using Eq. 4 yields

$$\tau_{\mathrm{syn}} = T_0 = T_u \exp[-(\beta/k)]. \tag{7}$$

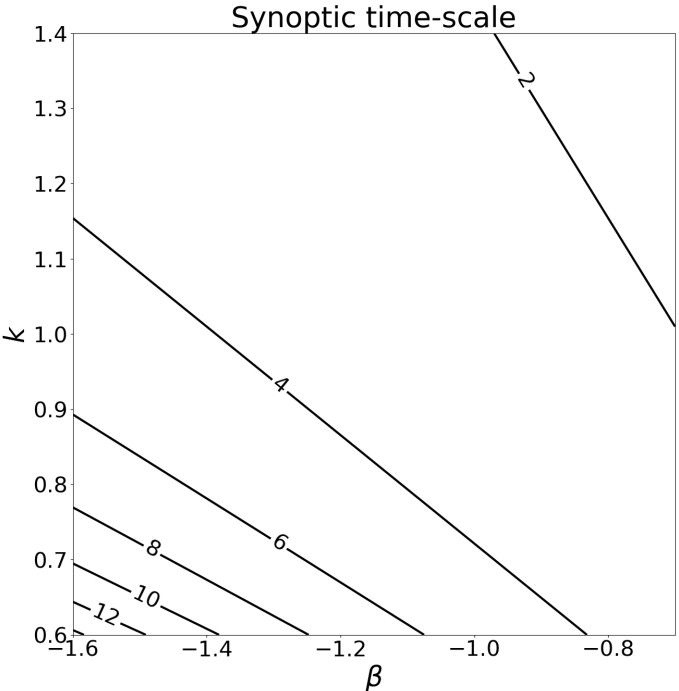

**Figure 2.** Time-scale for contrails to leave an ISSR (in hours). Small values of $\tau_{\mathrm{syn}}$ can be reached for small absolute values of $\beta$ and small values of $k$ and *vice versa*. Small values of $\tau_{\mathrm{syn}}$ indicate that the synoptic pathway of contrail dissolution may dominate.

### 3.2.2 Trajectory calculations and determination of synoptic time-scales

The synoptic time-scales are computed statistically. We consider air parcels at grid points with $PPC = 1$ and determine how
long they remain ice supersaturated during a series of WAWFOR forecast times up to 25 h. To this end, we perform trajectory
calculations, which are started one hour after the initialisation of the forecast. For each subsequent hour it is tested whether
an initially ice supersaturated air parcel is still supersaturated or not. For this decision, the $RHi$ value of the grid point where
the air parcels arrives after one hour (transported by the wind) is used. Air parcels that are still supersaturated are transported
further for the next hour, the others are no longer transported. This procedure is repeated until the last forecast step, $+25\,\mathrm{h}$.
Initially and at each subsequent time step the number of air parcels which are still supersaturated is recorded. These numbers





yield the survival function $S(t)$ (i.e. number of air parcels that are supersaturated at time step $t$, divided by the number of
initially supersaturated air parcels). The statistical analysis of this function provides the desired synoptic time-scale.

### 3.2.3   Case studies

In the following, various examples of the movement of areas with $PPC = 1$ at forecast initialisation are shown under different
weather conditions. The following three days are selected: 18/04/2024, 01/05/2024 and 24/05/2024. First, the synoptic weather
situation is briefly described for each day. For this, we show maps of the normalised geopotential height $Z^*$; the normalisation
procedure is described in Wilhelm et al. (2022). These maps show furthermore where the air moves upward (blue) or downward
(red) together with the ISSRs (stippling). Then, a series of plots show both the initial $PPC = 1$ region (red outline) and the
ISSRs one or more hours later (blue outline); additionally, the plots show with green colour all grid points where the initially
supersaturated air parcels (potential persistent contrails) have been transported to and where they survived so far. The green
area thus diminishes during the plot series. Finally, the survival function is plotted on the so-called Weibull paper, that is, in a
way such that a Weibull distribution appears as a straight line, from which we can determine slope and intercept.

In the first two examples, only the start and end times of the ISSR situations are shown. In the last example, some more steps
in between are also shown for illustration purposes.

**Case 1: April 18, 2024**

The weather situation on April 18, 2024 (12 UTC) and in the following hours is characterised by a trough at high altitudes
that begins over Sweden and extends over Germany, Italy, and as far as North Africa. This trough is closely linked to the
formation of high- and low-pressure regions on the ground. On the back of the trough, the isohypses run closer together and
converge. Due to the greater curvature of the isohypses at the southwestern tip of the trough axis, the wind is slowed down.
This leads to the air accumulating in front of the trough axis. In order to balance out the excess mass in front of the trough
axis, air masses are transported away. Since the tropopause is soon reached above, the air masses are preferentially transported
downwards towards the ground. This motion increases the air pressure on the ground and a high-pressure region is formed on
the back of the trough. On the front side of the trough, the curvature of the isohypses decreases as they diverge. This leads
to a lack of mass, which is compensated by air masses rising from the ground. This causes the air pressure on the ground to
fall and a low-pressure region is created. The large-scale sinking of air masses on the back of the trough and the rising in the
front of the trough can also be seen in Figure 3: there, on the back of the trough, mainly red areas (sinking) and on the front of
the trough mainly blue areas can be seen (rising). Consequently, ice supersaturation occurs on the front side of the trough, in
particular at its northeastern edge. It should also be emphasised that ice supersaturation occurs mainly at $Z^* \geq 0.98$.





(a) April 18, 2024, 12 UTC.

(b) April 18, 2024, 18 UTC.

(c) April 19, 2024, 00 UTC.

(d) April 19, 2024, 06 UTC.

(e) April 19, 2024, 12 UTC.

**Figure 3.** Synoptic charts for 18/19 April 2024. Shown are the normalised geopotential height (contours), the vertical velocity (in pressure coordinates) and the ISSRs (stippling) at 250 hPa.





In Figure 4 the $PPC = 1$ region at start time (April 18, 2024, from the forecast of 12 UTC + 1 h) is shown as the red contours in both panels. The blue contours show the ISSRs one hour later on the left and 24 hours later on the right panel. The green colour shows the initial $PPC = 1$ grid points that are still inside ISSRs, that is, they are all within the blue contours. After one hour there are still many green points, but 24 hours later only very few green points are still present.

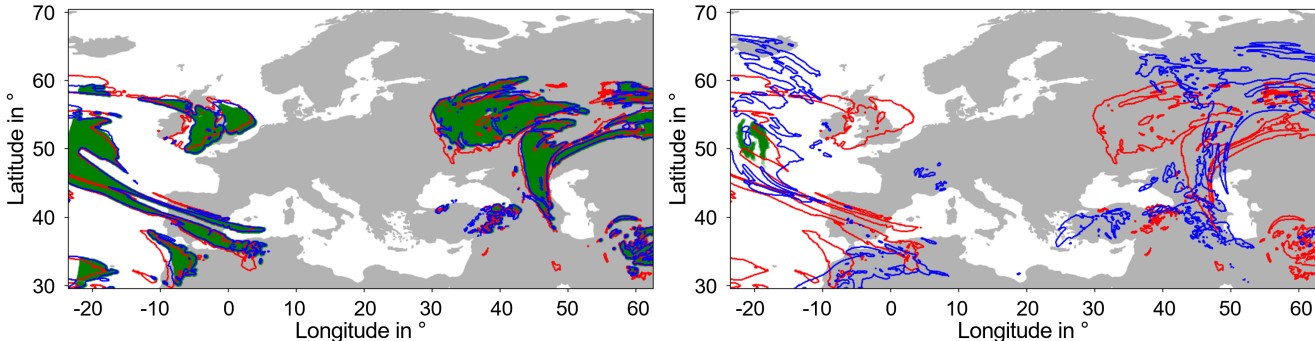

**Figure 4.** $PPC = 1$ regions (red contours) at the beginning (April 18, 2024, from the forecast of 12 UTC + 1 h) at 250 hPa in both panels, the ISSRs (blue contours) in the following hour (in the left panel) and 24 hours after the beginning (in the right panel). The green colour shows the initial $PPC = 1$ grid points that are still inside ISSRs.

Figure 5 shows the survival function of the number of grid points that were initially within $PPC = 1$ regions and that stayed continuously within ISSRs together with the best linear fit on Weibull paper. The fit can reproduce the curve very well and only deviates slightly towards the end for large values of $T$, where the curve begins to fluctuate due to noise. For this case, we determine a slope of $k = 0.9$ and an intercept $\beta = -1.1$. According to Figure 2, this implies a time-scale of approximately 4 hours.





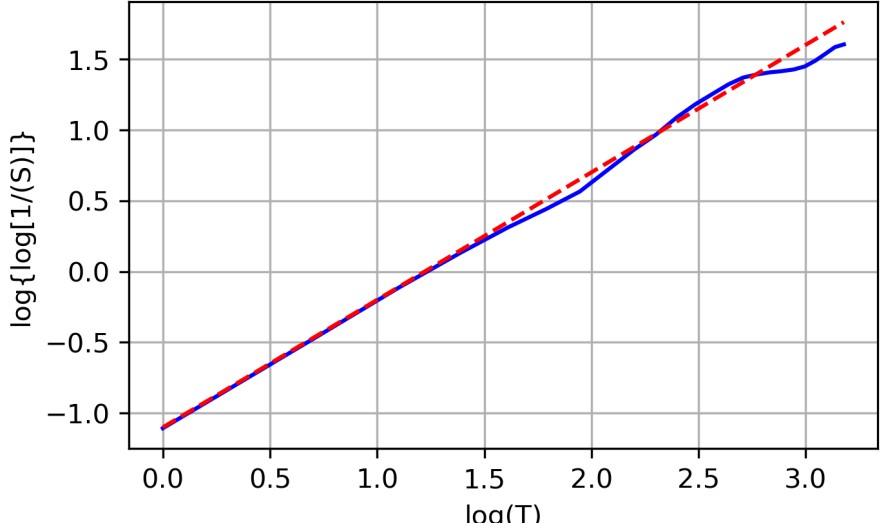

**Figure 5.** Survival function of the number of grid points that were initially within $PPC = 1$ regions and that survive by always being recorded within an ISSR in the following hours (blue) plotted on Weibull paper together with linear fits as dashed line: $g = 0.9 \cdot x - 1.10$ (red).

**Case 2: May 01, 2024**

The synoptic situation on May 01 and 02 is primarily determined by the wedge, which stretches from the Mediterranean
Sea off the coast of Libya and Egypt across Turkey, Poland, and Scandinavia to the northern coast of the United Kingdom (see Figure 6). Also in this case ice supersaturation mainly exists for $Z^* \geq 0.98$ in areas with upward air movement.



(a) May 01, 2024, 12 UTC.

(b) May 01, 2024, 18 UTC.

(c) May 02, 2024, 00 UTC.

(d) May 02, 2024, 06 UTC.

(e) May 02, 2024, 12 UTC.

**Figure 6.** Synoptic charts for 01/02 May 2024. Shown are the normalised geopotential height (contours), the vertical velocity (in pressure coordinates) and the ISSRs (stippling) at 250 hPa.



Figure 10 shows the $PPC = 1$ region at start time (May 01, 2024, from the forecast of 12 UTC + 1 h) as the red contours in both panels. The blue contours show the ISSRs one hour later on the left and 24 hours later on the right panel. The green colour shows the initial $PPC = 1$ grid points that are still inside ISSRs. After one hour there are still many green points, but after 24 hours almost all have vanished.

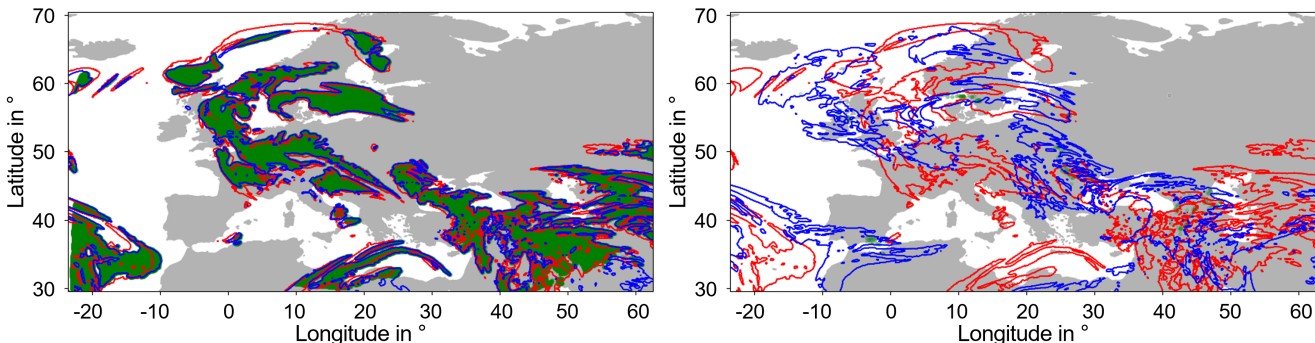

**Figure 7.** $PPC = 1$ regions (red contours) at the beginning (May 01, 2024 from the forecast of 12 UTC + 1 h) at 250 hPa in both panels, the ISSRs (blue contours) on the following hour (in the left panel) and 24 hours after the beginning (in the right panel). Green colour shows the initial $PPC = 1$ grid points that are still inside ISSRs.

The survival function for this case can excellently be fitted with a Weibull distribution with a slope of $k = 0.92$ and an intercept $\beta = -1.09$, see Figure 8. The synoptic time-scale is thus again about 4 h.



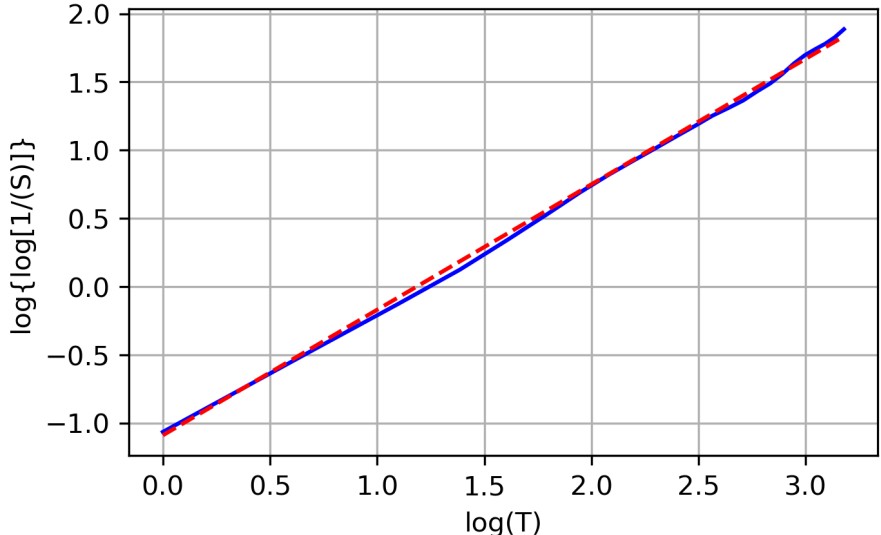

**Figure 8.** Survival function of the number of grid points that were initially within $PPC = 1$ regions and that survive by always being recorded within an ISSR in the following hours (blue) plotted on Weibull paper together with linear fits as dashed line: $g = 0.92 \cdot x - 1.09$ (red).

**Case 3: May 24, 2024**

This situation is characterised by exclusively high values of the normalised geopotential height and also large areas with ice supersaturation over Scandinavia, Russia, Central Europe, Iran, Libya, and Egypt, which occurs mainly in raising air at $Z^* > 1.0$.





**Figure 9.** Synoptic charts for 24/25 May 2024. Shown are the normalised geopotential height (contours), the vertical velocity (in pressure coordinates) and the ISSRs (stippling) at 250 hPa.



Figure 10 shows the evolution of the surviving initial contrail grid points in more detail, that is, with more intermediate time steps. The $PPC = 1$ region at start time (May 24, 2024, from the forecast of 12 UTC + 1 h) is again shown as the red contours in all panels. The blue contours show the ISSRs 1 hour (a), 3 hours (b), 5 hours (c), 10 hours (d), 15 hours (e) and 24 hours (f) later. The green colour shows as usual the initial $PPC = 1$ grid points that are still inside ISSRs at the respective time steps. The number of green points decreases continuously over time.

**Figure 10.** All panels show the $PPC = 1$ regions at the beginning (May 24, 2024 from the forecast of 12 UTC + 1 h) at 250 hPa as red contours. The six different panels indicate ISSRs from a selection of the following hours in blue (horizontally from top left to bottom right: ISSRs 1 h later at 12 UTC + 2 hrs; ISSRs 3 hrs later; ISSRs 5 hrs later; ISSRs 10 hrs later; ISSRs 15 hrs later and ISSRs 24 hrs later). Grid points that were within ice supersaturation in every subsequent hour since the beginning are marked in green.





Figure 11 shows the survival function for this case. The best Weibull fit for this case has a slope of $k = 1.02$ and an intercept $\beta = -1.30$. This is nearly an exponential distribution (which would have $k = 1$). The corresponding time-scale is again about 4 hours.

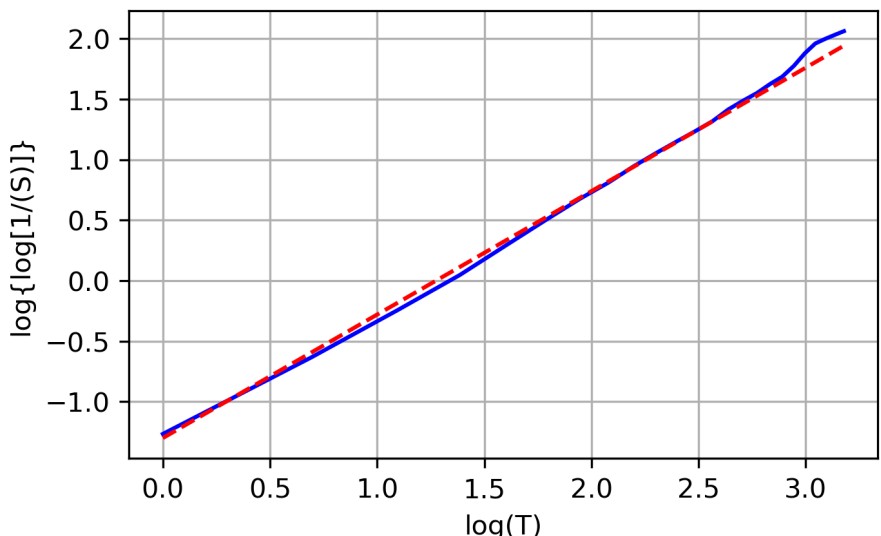

**Figure 11.** Survival function of the number of grid points that were initially within $PPC = 1$ regions and that survive by always being recorded within an ISSR in the following hours (blue) plotted on Weibull paper together with linear fits as dashed line: $g = 1.02 \cdot x - 1.30$ (red).

## 4 Discussion

### 4.1 Time-scales and lifetimes

It is important to distinguish time-scales from lifetimes. The lifetime of a contrail can be defined as the period from its formation until it can no longer be seen or until some other threshold is underrun. In principle, such a period can precisely be determined, although this is in most cases not required. In contrast, a time-scale is rather understood as a rough estimate than a precise value. It is the time that it takes for a considerable (another vague word) change to a system, for instance, an e-folding time. (This does not exclude the possibility of giving precise e-folding times, for instance for radioactive decay). If we take a time-scale as an e-folding time, for instance, for the loss of ice mass, a corresponding lifetime would be twice or three times as long (as $e^{-2} \approx 0.14$ and $e^{-3} \approx 0.05$). One should also note that factors of the order one are unimportant for time-scale considerations. For instance, the shrinking of ice crystals can be considered via the reduction of its mass or its radius. Although the same process is considered, the corresponding time-scales differ by approximately a factor of three (for spherical crystals).





It follows from these considerations that it is not necessary to put very much effort into the numerical determination of the synoptic time-scale. In our view, the calculation of trajectories in the horizontal dimension with a simple Euler-forward method and the application of the nearest neighbour approach to map relative humidity values to trajectory locations after each time
step is sufficient for the estimation of a time-scale.

However, some systematic errors cannot be excluded. When calculating the synoptic time-scale for contrails to leave their parent ISSR, we only have 1-hour time steps. One cannot determine whether an air parcel resided always inside the ISSR during this hour. Air parcels at the edge of an ISSR could leave and re-enter it within one hour, or even re-enter another ISSR that might be nearby or appear as a new ISSR. This can lead to errors, in particular when ISSRs in the forecast have small-scale
structures such that their perimeter-to-area ratio is large. The problem should be small for large continuous ISSRs with little small-scale structure (in the forecast).

At the start of the trajectory calculations, the ISSRs are already present and have already an age, that can exceed a day. One might believe that this leads to an underestimation of the synoptic time-scale, but we do not think so. Had we tried to determine a lifetime of ISSRs, then this could be a source of error. But here we are concerned with the time it needs for an air
parcel to leave an ISSR due to the differences in the motion of the wind and the motion of the ISSR (Hofer and Gierens, 2025). We do not see how this time-scale could be biased by the fact that the ISSRs have already some age at the beginning of our calculations.

## 4.2 Comparison with results from the literature

The results obtained in our case studies all gave Weibull-type survival functions, with parameters $\beta$ slightly below $-1$ and $k$
between $0.9$ and about $1$. The corresponding synoptic time-scales are close to 4 hours in all three cases. We can compare these results with results from the literature.

Let us first mention that contrail lifetimes obtained from a study using satellite-tracking of contrails (Gierens and Vázquez-Navarro, 2018) gave a mean lifetime of $3.7\,\mathrm{h}$ which is consistent with both the synoptic and the sedimentation time-scales derived in this study (although the comparison is a little off since the mean of a Weibull distribution differs from its scale
parameter). Unfortunately, the satellite-tracking cannot distinguish whether a contrail disappeared due to sedimentation of its crystals or due to sublimation in dry air, in particular, as the late contrail evolution can often not be observed in satellite images due to vanishing contrast. To determine the mean lifetimes in the satellite study, the authors applied statistical arguments and modelling, based on the Weibull distribution, which provided a good fit to the observed survival function. The parameters in that study lead to a quite short time-scale of about $0.6\,\mathrm{h}$, a consequence and indication of the fact that only a part of the contrail
lifetime can be observed via satellite-tracking.

An immediate comparison is possible with results from trajectory calculations presented by Dietz (2012). He uses NWP data obtained from the operational version of the COSMO-Model (the model that was operational at DWD at that time, including a one-moment microphysics scheme) and from a non-operational version with a 2-moment scheme (Seifert et al., 2012) and applies fits of Weibull distributions to the survival functions as well. Additionally, two versions of trajectories are calculated:
trajectories from the model data (that is affected by all processes implemented in the model) and a version where the effects





of microphysics are switched off by conserving the initial absolute humidity in the calculation (which in reality is reduced by cloud formation) to obtain a "virtual" relative humidity. The following time-scales are determined ($k$-values in brackets):

- 1-moment scheme: $3.0$ hours ($k = 0.9$);

- 2-moment scheme: $3.8$ hours ($k = 0.75$);

- 1-moment scheme, virtual RHi: $4.0$ hours ($k = 0.75$);

- 2-moment scheme, virtual RHi: $5.6$ hours ($k = 0.875$).

Two observations are immediately possible: first, the 2-moment scheme leads to longer time-scales, and second, the virtual humidity fields have longer time-scales than the real humidity fields. We think this can be explained quite easily. The 2-moment scheme allows higher ice supersaturation to be obtained in the NWP model than the 1-moment scheme. Thus, ISSRs in the 2-moment schemes have on average higher maximum humidities and it takes longer for them to dry out than for the 1-moment scheme. Switching off microphysics, that is in particular switching off sedimentation of ice crystals, leads again to longer time-scales in both model versions. These two time-scales are probably those that are most directly comparable to the ones that we derived here. Indeed, as our trajectories are based on ICON equipped with a 1-moment model, the synoptic time-scales are equal: 4 hours. We note, however, that our $k$-values are slightly larger than those obtained by Dietz (2012). This difference might result from methodical differences between our study and that of Dietz (for instance, he used 3-dimensional trajectories, calculated with a Runge-Kutta method; our approach was simpler with a 2-dimensional Euler-forward method).

Numerical simulations of contrails, either with cloud-resolving models (eg. Unterstrasser and Gierens, 2010a, b; Lewellen, 2014) or with global circulation models (eg. Bier et al., 2017), provide information on contrail dissolution processes as well. Cloud-resolving simulations usually assume constant synoptic conditions such that contrail dissolution is then only possible via microphysical processes. But it is not only sedimentation that occurs then. Depositional growth of ice crystals reduces the supersaturation in the contrail-containing ISSRs, but it does not lead to subsaturation. Thus, crystal loss in this case is still dominated by sedimentation. Unterstrasser and Gierens (2010a) observe in their simulations fallstreaks developing after $6500\,\mathrm{s}$ ($1.8\,\mathrm{h}$), but these consist only of very few large crystals. Most of the contrails cease to grow in mass after 3-4 hours when sedimentation and crystal growth (in a constant supersaturation field) balance. Without synoptic evolution, the total extinction first grows and then, after about 3 h, it stagnates or begins to decline, which is due to sedimentation. The authors consider the 3 hours an intrinsic time-scale for contrails, where "intrinsic" means that it is determined by microphysics rather than synoptic evolution. All the sensitivity studies in Unterstrasser and Gierens (2010b) show the same 3 hours intrinsic time-scale. Lewellen (2014) often finds contrails with lifetimes exceeding 10 h. This may seem quite high, but it is within the range of time-scales derived here. Synoptic changes that would lead a contrail into sub-saturated environments are not simulated, and the height of the supersaturated layer is assumed to range from 500 to $1500\,\mathrm{m}$. As the author simulates contrails up to their complete demise, contrail lifetimes can be quite long. For comparison, with a height of 1000 m, the time-scales shown in our Figure 1 must be doubled. Since these are e-folding times, the total lifetimes can easily be two or three times as long. Thus we find no contradiction between the present and Lewellen's results.



The simulation with a global circulation model, that for the first time (to our knowledge) introduced the distinction between
the microphysical and the synoptic pathway to contrail dissolution, was that of Bier et al. (2017). These authors also mentioned
precipitation (aggregation of ice crystals to snow-flakes which fall) and the mixing with ambient natural cirrus (or replacement
of contrail cirrus by natural cirrus) as contrail terminating processes, however with considerably smaller effect than sedimenta-
tion and synoptic drying. In the study, large contrail clusters are formed on several days and their evolution is observed. In the
eight considered cases, sedimentation dominates contrail dissolution in three cases, and synoptic drying in two. The remaining
three cases are transition cases (probably cases with $\tau_{sed} \approx \tau_{syn}$). Although the lifetimes of a contrail cluster are not simply
comparable to the time-scales considered here, we can see from their figures that several properties of the clusters change at a
high rate during the first few hours, but then, after about 5 to 8 hours, further changes are weak. All cases show this behaviour
and we think, this suggests that both time-scales should be of the order of 5-8 hours, consistent with our results. Bier et al.
(2017) study as well the effect of a reduction of the initial ice crystal number in the contrails, e.g. due to alternative fuels. This
leads to larger ice crystal masses and thus, according to Eq. 3 to a shorter sedimentation time-scale. Accordingly, all simula-
tions with reduced ice crystal numbers show shorter contrail lifetimes. Interestingly, the contrail life times are also reduced in
the dynamically controlled cases. We think the explanation for this is probably that in the simulations both processes occur
simultaneously and sedimentation is active in dynamically controlled cases as well.

As both processes, sedimentation and synoptic evolution, occur simultaneously in nature, the combined time-scale is half
the harmonic mean of the two time-scales, because:

$$
\begin{aligned}
\frac{1}{\tau} &= \frac{1}{\tau_{sed}} + \frac{1}{\tau_{syn}}, \qquad \text{thus} \\
\tau &= \frac{\tau_{sed}\,\tau_{syn}}{\tau_{sed} + \tau_{syn}}
\end{aligned}
\tag{8}
$$





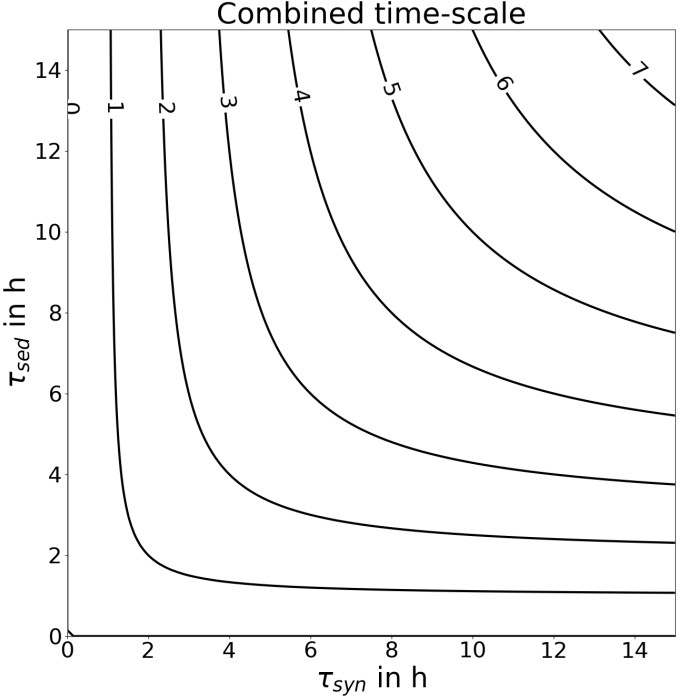

**Figure 12.** The combined time-scale of $\tau_{\mathrm{sed}}$ and $\tau_{\mathrm{syn}}$ in hours calculated according to Equation 8.

The combined time-scale is thus shorter than the two time-scales in separation, see Figure 12. It is also shorter than the smaller one of the two time-scales. Even if both time-scales would be ten hours, the combined one is only five hours, which is plausible, since two processes act simultaneously to dissolve the contrail. This explains probably why the satellite observations give shorter time-scales than the cloud-resolving simulations which typically assume constant synoptic conditions and horizontally periodic boundary conditions.

### 4.3 ISSRs in the synoptic situation

The synoptic weather charts shown to describe the three cases display the normalised geopotential height, $Z^*$, instead of the geopotential *per se*. This quantity has been introduced by Wilhelm et al. (2022) to make variations of this field comparable, i.e. to bring them on a common scale, for different pressure levels. This scale ranges from about $0.9$ to about $1.1$, independent of pressure in the upper troposphere. Wilhelm et al. (2022) already pointed out that regions where persistent contrails can form are characterised by high values of $Z^*$. This finding is corroborated by the current results: we find ISSRs (stippling) mainly where $Z^* > 0.98$. In contrast, ISSRs are rarely found where $Z^* < 0.98$. Of course, ISSRs and regions with $Z^* > 0.98$ are not identical. That implies, that for contrail avoidance it is still necessary to predict ice supersaturation. But the fact the ISSRs are hardly present in regions with $Z^* < 0.98$ renders the prediction work easier. In such regions it is not necessary to think about contrail prevention, they will hardly persist if they form at all.



This result could only be obtained by normalising the geopotential. Had we not normalised it, the boundary that corresponds to $Z^* = 0.98$ would appear at different heights in different pressure altitudes and it would be difficult to notice that there is
such a boundary at all.

## 5   Conclusion

Contrails are dissolved mainly by the following processes:

– Sedimentation of the ice crystals into lower levels that are sub-saturated;

– The (horizontal) wind blows the ice crystals out of the parent ISSRs;

– Large-scale subsidence diminishes supersaturation down to subsaturation.

The first of these processes can be characterized by a sedimentation time-scale, $\tau_{\mathrm{sed}}$, which is proportional to the height of the supersaturated layer and which depends weaker than linearly on the mean mass of the contrail ice crystals. Typical values are a couple of hours.

The other two processes can together be characterized by a synoptic time-scale, $\tau_{\mathrm{syn}}$, which does not depend on character-
istics of the contrail or its ice crystals but only on the large-scale synoptic situation and in particular the relative motion of the parent ISSR and the local wind (see, Hofer and Gierens, 2025). The synoptic time-scale is also on the order of a couple of hours.

The fact that both time-scales are similar may explain that it is unknown whether the microphysical or the synoptic pathway dominates in contrail dissolution. But it can as well imply that often both pathways are of similar importance.
As other contrail-removing processes are of minor importance, it is the combination of the two time-scales that characterises the lifetime of contrails. This combination is half the harmonic mean of the two time-scales in separation, which is always less than the smaller of the two time-scales. The time-scales in the current paper are defined as e-folding times. Total lifetimes are perhaps rather 2 or three e-folding times.

Cloud-resolving models of contrails usually assume constant synoptic conditions and, applying periodic boundary condi-
tions, effectively assume horizontally infinitely extended ISSRs. Thus, the synoptic time-scale is effectively infinite. Contrail lifetimes in these models are often quite long ($> 10 \, \mathrm{h}$). In contrast, contrail simulations in global circulation models where both pathways are effective yield shorter contrail lifetimes (say, about $5 - 8 \, \mathrm{h}$). Both pathways are also effective in the real world. Contrail tracking studies using satellite imagery find thus short lifetimes of the order $4 \, \mathrm{h}$.

The sedimentation time-scale of a contrail can be diminished if alternative fuels with reduced soot emission (by number) are
applied, as long as the reduction does not lead into the so-called soot-poor regime (Kärcher and Yu, 2009). Less soot implies less but larger ice crystals (i.e. with higher mean mass). Thus, the microphysical time-scale can be reduced by technical means, whereas the synoptic time-scale is given by the weather situation. To be an effective means of contrail mitigation, the resulting sedimentation time-scale must be smaller than the synoptic time-scale. In order to determine this in the flight planning phase trajectory calculations would be necessary.





Finally, a side result of this study is that contrails will hardly persist in regions where the normalised geopotential height, as defined by Wilhelm et al. (2022), is less than $0.98$. This simple boundary can easily be calculated for the upper troposphere and we recommend strongly that aviation weather forecasts use normalised geopotential height on their synoptic charts because this allows flight planners to see immediately where contrail prevention actions are not necessary.

*Code availability.* Python codes can be shared on request.

*Data availability.*

*Author contributions.* This paper is part of SH's PhD thesis. SH wrote the codes, ran the calculations, analysed the results and produced the figures. KG supervises her research. Both authors discussed the methods and results and wrote the paper.

*Competing interests.* The authors declare no competing interests.

*Acknowledgements.* This research contributes to and is supported by the project D-KULT, Demonstrator Klimafreundliche Luftfahrt (Förder-
kennzeichen 20M2111A), within the Luftfahrtforschungsprogramm LuFo VI of the German Bundesministerium für Wirtschaft und Kli-maschutz. This work used resources of the Deutsches Klimarechenzentrum (DKRZ) granted by its Scientific Steering Committee (WLA) under project ID bd1357. The authors would like to thank Simon Kirschler for his thorough reading and commenting a draft manuscript.



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
