# Peer review of "Synoptic and microphysical lifetime constraints for contrails"

_EGUsphere, 2025_

## Referee Comment (RC1)

**Comments on "Synoptic and microphysical lifetime constraints for contrails" by Sina Maria Hofer and Klaus Martin Gierens**
**https://doi.org/10.5194/egusphere-2025-326**

The objective of this study is to identify the typical synoptic and microphysical time-scales that constrain contrail lifetime and inform the targeted use of alternative fuels.
The manuscript uses hourly aviation weather forecast data based on ICON simulations from the German Weather Service. The model output of temperature and relative humidity is used in combination with the Schmidt-Appleman-criterion to estimate the potential for contrail formation, the persistence of contrails, and their tempo-spatial evolution over a 24 hour period. Information about the wind speed and wind direction is used to infer the advection of the detected ice supersaturated regions (ISSRs)  and to calculate parcel trajectories.
The analysis is based on a total of three days in April and Mai 2024.
Based on this analysis, the authors find that in most of the analyzed cases, the synoptic and microphsyical time-scale are of similar length and argue that they are likely to be equally important. The authors further state that an efficient application of alternative fuels requires the microphysical time-scale to be smaller than the synoptic times-scale.

I do have some major comments, which mostly address the structure of Section 4.1, which could be clearer and in a better order. In general, a more concise writing, for example by avoiding jargon, can aid the understanding of the entire paper.
Besides the major comments there are a number of minor comments, mostly related to style, which does not follow the ACP submission guidelines. These could easily be avoided. Some of the comments are of a suggestive nature and could be considered by the authors. Other comments ask for more precise wording.

After the authors have adequately addressed the major and minor comments below, the manuscript may be considered for publication in ACP.

**Major comments:**

Section 4.1 is, in my opinion, unnecessary long. The authors discuss about potential definitions. This section could be much shorter and to the point by defining their time-scales. But this is already done in Section 3.1. Why not include Section 4.1 in Section 3 and define your time-scales and lifetimes there?

L216: Regarding the comparison between observations and models. Isn't this a more fundamental question of several components? In both, the measurements and in the simulations, you have to define a cloud by some kind of threshold (see also minor comments L178ff). What is detected as a cloud depends strongly on the definition of a cloud, the threshold, and how and by what means the cloud is detected.
Second, the authors identify ISSRs, i.e., regions that **allow** the formation of persistent contrails. Contrarily, observing contrails from satellite, identifies and tracks contrails that **actually formed**. It would be good if the authors could comment on this.
Third, what is the effect of your threshold of $RHi>93\%$ for ice-supersaturation and the size of the ISSRs? To answer this question, the authors could vary the threshold for ice-supersaturation and check the robustness of the results.

In the second last section of the summary, the authors state that for targeted use of alternative fuels, trajectory calculations are required. If I am not mistaken, this would be required for several flight levels and over larger areas, where aircraft are flying and potentially interact with ISSRs. This can be computationally challenging. Perhaps the authors could provide a few comments on the following points:
- My interpretation of this approach is that it is a near real-time avoidance strategy that may be limited by airspace restrictions and air traffic control, as flight plans and routes are well defined days to months in advance. This may change in the future, but is difficult to apply today.
- Trajectory calculations are time consuming, requiring the calculation to be started well in advance of the flight day in order to adjust the flight path (if at all possible on short notice).
- The approach relies on the accuracy of the underlying weather model to predict the exact location of the ISSRs several days in advance, since it must include the time for the trajectory calculation plus the time to adjust the flight.
The authors could partially answer this by providing the times they needed to perform the calculations.

**Minor comments:**

L20: " in air with sub-saturated water vapor" Please reconsider this phrase. Water vapor cannot be subsaturated by itself. You are probably referring to the air, which can be subsaturated.

L27: Please clarify what you mean by "All of these effects should become stronger with increasing contrail lifetime". If one considers constant illumination conditions but a thinning of the contrail with time, as you show in the manuscript, then the radiative effect would decrease even though the contrail can be considered persistent. By rephrasing this sentence, you might better explain your actual intent.

L36: "dominate" instead of "dominates"?

L38. "Fewer" instead of "less"?

L56: "data inform". Data is singular, so I think it should be "data informs". You might consider rephrasing it, since data itself cannot inform.  You could write something  like: "based on the data, ...."

Fig1 and subsection: Please specify the unit of $r_0$ on the $y$-axis. Furthermore, if the isolines represent the time-scale, why did the authors choose an example that is "not included" in the figure. It would be more intuitive and easier for the reader to choose an example with a time-scale of 4.5h

or 6h, since these lines are included. Conversely, you could add the 5h isoline. Does the word "deep" refer to the geometric thickness of the cloud?

Eq.6 Please check that the double "ln" is correct.

L135-136: Instead of "transported downwards to the ground" the term "subsidence" could be used, which is shorter and more common in this context and refers to large-scale sinking air masses.

Fig4. You might consider adding (a) and (b) to the panels to make them more easily referenced in the caption and text.

L146: Instead of qualitative measures, it would be better to quantify and provide absolute numbers and/or percentages of how many air parcels remained ice-supersaturated.

Fig5.: Equation 5 uses "ln", which has a base of e. However, the *y*-axis in Fig.5 uses "log", which usually has a base of 10. Please check for consistency.

L154: Here and elsewhere in the text, e.g., L67: Please check the date format here, in the caption of Figure6, and compare with the ACP submission guidelines.
Copied from the ACP submission guidelines:
"Date and time: 25 July 2007 (dd month yyyy), 15:17:02 (hh:mm:ss). Often it is necessary to specify the time if referring to local time or universal time coordinated. This can be done by adding "LT" or "UTC", respectively. If needed when referring to years, CE (common era) and BCE (before the common era) should be used instead of AD and BC since CE and BCE are more appropriate in interfaith dialogue and science."

L159-161: You might consider quantifying "many" and "almost all".

L161: Please quantify "excellently". For example, by providing an r-squared for the fit.

L178-179: The phrase "...can no longer be seen…" is imprecise. "Visibility" of contrails and water vapor is strongly dependent on wavelength. Thus, the authors would need to provide a wavelength range in which the observation takes place. Please see Driver et al. 2025, where they state that there is no clear threshold (https://amt.copernicus.org/articles/18/1115/2025/). It might be better to provide a threshold to describe the presence of contrails, such as, ice mass.

L179: "In principle, such a period can precisely be determined,…" I doubt that this is an easy task. As mentioned before, it is almost a philosophical question what counts as a cloud or water vapor. Clouds that are optically thin at solar wavelengths can still be very effective at absorbing radiation in the thermal infrared. To make such a judgment, a threshold must be used. Since, to my knowledge, there is no consensus on such a threshold, the authors may define their own threshold based on reasonable judgment.

L181: "…  (another vague word) …" What is the authors intention of this insert? The first paragraph of this section could be omitted, or at least shortened, by providing a clear definition of the time-scale as understood and defined by the authors.

L198: "One might believe…" and ".. but we do not think so" are expressions that should be avoided in scientific publications. Hypotheses and conclusions are based on reasoning and facts. Please consider rephrasing this sentence.

L216-219: Please check the grammar in this sentence

L219-222: Please consider rephrasing this sentence. What you probably mean: There are two types of trajectory calculations: 1) following the air parcel in normal mode, and, as I understand it, 2) following the air parcel but with precipitation / sedimentation and condensation switched off. This is mentioned later in L231, but would fit here.

L219-222: The term of "virtual" humidity may be an unfortunate choice, as it may be confused or associated with other metrics such as virtual temperature. You might consider calling it along the lines of "idealized simulations". However, this is just a personal impression and I would not insist on a change.

L225 and 226, RHi should be  "$RH_i$" for consistency

L249ff: the word "height" is used, which is misleading. I assume you mean cirrus / cloud geometric thickness, right?

L312: The authors please better explain what they mean by "real world". A better term might be "in nature"?

L315: "soor-poor-regime" is mentioned here for the first time. The authors could mention this earlier when introducing the targeted use of alternative fuels.

For the entire text:

- If I am not mistaken, the first appearance of the figures in the text follows: 3, 4, 5, 2, 6, 10, 8, 11, 1, and 12. Consequently, the inclusion of the figures should be reordered. Figures 9 and 11 are not referenced anywhere in the text. Either refer to them in the text or, if they are not needed, remove Figures 9 and 11.

- The authors are inconsistent in their use of "Section", "section", and "Sec."
  Copied from the ACP submission guidelines: "The abbreviation "Sect." should be used when it appears in the running text and should be followed by a number unless it is at the beginning of a sentence."

- "Figure" is written at the beginning of a sentence and abbreviated within a running sentence. Copied from ACP submission guidelines: "The abbreviation "Fig." should be used when it appears in running text and should be followed by a number unless it comes at the beginning of a sentence, e.g.: "The results are depicted in Fig. 5. Figure 9 reveals that...".

- Please be consistent in your use of the serial comma if you choose to use it.

- Please be consistent in your use of units. Sometimes "hours" is used and sometimes "hrs". Check the ACP guidelines.

- Variables are usually in italics, while subscripts are in normal font. For example: instead of $RH_i$ write $RH_\text{i}$

---

## Referee Comment (RC2)

Review of „Synoptic and microphysical lifetime constraints for contrails"
by Sina Hofer and Klaus Greens submitted to EGUsphere

The submitted paper by Hofer and Gierens analyses the spatial and temporal development of ice-supersaturated regions (ISSRs) in connection with large-scale weather patterns over Central Europe. The aim is to derive lifetimes for contrails. Therefore, two different lifetimes are considered, which are based on different processes. First, the microphysical lifetime, which is primarily determined by the sedimentation of the ice crystals. The second lifetime approach focusses on the synoptic time scale, whereby the degradation of ice supersaturation through synoptic processes and the departure of air parcels from the ISSR are considered.

The paper addresses an important issue for reducing the negative effect of aviation. However, in my opinion, the paper could be written more precisely and thus more briefly, as I will suggest below. I therefore recommend publishing the paper after the major revisions.

**Case studies**
The choice of the three case studies is not clear to me. I understand that they are supposed to describe three different synoptic situations. However, these are situations within a period of 5 weeks in 2024 during the transition from spring to summer over continental Europe. This raises the question what results could be expected for other seasons?

The description of the case studies in text form compared to the number of images seems lopsided to me. This means that the reader has to leaf through a lot, but learns little new. Perhaps it would make more sense to discuss one case study in more detail and only draw attention to clear differences for the other case studies with fewer illustrations and text?

The synoptic charts (Figs, 3,.6,9) are very small but full of detailed information, which does not always differ from time to time. I would therefore suggest greatly reducing the number of subplots. To describe the weather situation, only one figure might be sufficient.

Fig. 5,8,11 could be summarised as a triple plot and discussed together. Make sure that the y-axis is always identical. Currently, all y-axes are slightly different.

Fig. 5: It is mentioned in the text that the „fit only deviates slightly towards […] large values of T […] due to noise". Why does noise play a role for large values of T? Also: what is meant by noise here in the first place?

Finally, the conclusions drawn relate, among other things, to the three case studies, which were only conducted over Europe in April and May 2024. The conclusions drawn therefore seem a little too generalised to me. Shouldn't other time scales be expected over the Atlantic or Pacific with stronger links to the tropics? Perhaps the authors could discuss this in more detail.

**Other remarks**
L20: Perhaps splitting hairs, but can „water vapor" be subsaturated

L24f: Three very short sentences in a row. Maybe connect the last two sentences into one?

L45: „…it is not yet known which pathway dominates." Why should one process dominate?

L62: how robust are the results on the ISSR definition of RHi>93%?

L65f: What is the vertical resolution of the underlying model?

Eq (6): Parenthesis in equation is missing.
Also, here you refer to ln, later in Fig. 5,8,11 you refer to log? Do you mean the same? Please clarify.

Sec. 3.2.1: I find it misleading to denote the time with a capital „T" and would suggest a small „t"

Sec. 3.2.3: Regarding Z*. It would help the reader if the brief description of Z* is provided already here and not in Sec. 4.3

L130: what is „high altitudes"?

L165: what is meant with „exclusively"?

Fig. 10: labeling missing ((a) to (f))

Sec 4.1 feels out of place here. The discussion about the difference between lifetime and time-scales would be better placed in the introduction. The discussion of trajectory calculation, on the other hand, would also be useful in the methods section. Also: why are methods and results combined in one section?

L 284f: what does „mainly" and „rarely" quantitative mean?

L 286: Will the prediction be easier if the normalised geopotential is calculated first in order to narrow down the calculation of the ISSR? This introduces a further work step.

Generally, please refrain from using statements such as 'we think' or 'eplains probably' as this leaves a lot of room for speculation.

---

## Author Comment (AC1)

**Replies**

Sina Hofer and Klaus Gierens

**Answers to Reviewers**

We thank the reviewers for the comments. For convenience, we repeat the comments and then give our replies, which are printed in italics.

**Review 1:**

The submitted paper by Hofer and Gierens analyses the spatial and temporal development of ice-supersaturated regions (ISSRs) in connection with large-scale weather patterns over Central Europe. The aim is to derive lifetimes for contrails. Therefore, two different lifetimes are considered, which are based on different processes. First, the microphysical lifetime, which is primarily determined by the sedimentation of the ice crystals. The second lifetime approach focusses on the synoptic time scale, whereby the degradation of ice supersaturation through synoptic processes and the departure of air parcels from the ISSR are considered.

The paper addresses an important issue for reducing the negative effect of aviation. However, in my opinion, the paper could be written more precisely and thus more briefly, as I will suggest below. I therefore recommend publishing the paper after the major revisions.

**Case studies:**

The choice of the three case studies is not clear to me. I understand that they are supposed to describe three different synoptic situations. However, these are situations within a period of 5 weeks in 2024 during the transition from spring to summer over continental Europe. This raises the question what results could be expected for other seasons?

*REPLY: It is a pity that the WAWFOR-Klima data changed after May 2024, such that more data could not be analysed without inconsistency. But, the comment is of course justified, and indeed it is quite possible and plausible that the synoptic time-scales vary from season to season and from region to region. We provide now a short discussion of this as the first subsection in the Discussion section. Additionally, the conclusions are modified, following your advice below.*

*The description of the case studies in text form compared to the number of images seems lopsided to me. This means that the reader has to leaf through a lot, but learns little new. Perhaps it would make more sense to discuss one case study in more detail and only draw attention to clear differences for the other case studies with fewer illustrations and text?*
*REPLY: We agree. Indeed the weather changes slowly enough that it may suffice to illustrate each case with a single weather chart. Otherwise, we think, we have already fulfilled the reviewer's suggestion. While case 1 is described at some length, the two others are described in a few lines.*

*The synoptic charts (Figs, 3,.6,9) are very small but full of detailed information, which does not always differ from time to time. I would therefore suggest greatly reducing the number of subplots. To describe the weather situation, only one figure might be sufficient.*
*REPLY: We agree, see above.*

*Fig. 5,8,11 could be summarised as a triple plot and discussed together. Make sure that the y-axis is always identical. Currently, all y-axes are slightly different.*

*REPLY: Indeed, these 6 lines could even go into one figure, with different colours, of course. However, having all these plots at a single place would lead to the inconvenience for the reader that it becomes necessary to leaf forward and backward while reading. We think this is not nice and we prefer to leave the figures separated.*

*Fig. 5: It is mentioned in the text that the „fit only deviates slightly towards [. . . ] large values of T [. . . ] due to noise". Why does noise play a role for large values of T? Also: what is meant by noise here in the first place?*
*REPLY: Initially (small $T$), there are many air parcels and the statistics is good. The longer $T$ gets, the fewer air parcels remain and the statistics gets worse. This is usually described as noise in the data. We reformulate the sentence as "where the statistics gets worse because the number of surviving air parcels gets smaller and smaller with increasing $T$."*

*Finally, the conclusions drawn relate, among other things, to the three case studies, which were only conducted over Europe in April and May 2024. The conclusions drawn therefore seem a little too generalised to me. Shouldn't other time scales be expected over the Atlantic or Pacific with stronger links to the tropics? Perhaps the authors could discuss this in more detail.*
*REPLY: We agree. We have thus added the new subsection in the discussion section, see above. Also, it is not sure whether the relation between ISSRs and the normalised geopotential is useful in other regions of the world. Eventually, this relation has been found in aircraft data obtained in the northern mid-latitudes (Wilhelm et al., 2022). Other conclusions are general. For instance, the sedimentation time-scale is independent of the case studies. Anyway, we add cautionary notes to the conclusion which clearly state, the the relations found in the present paper may be valid only over Europe and might be different in other regions.*

**Other remarks:**

*L20: Perhaps splitting hairs, but can „water vapor" be subsaturated*
*REPLY: It is good that this comment is formulated as a question. The answer is "yes". To be clear: only the water vapour can be subsaturated. In spite of this, we also used this common phrase (subsaturated air) at several locations in the paper. In order to avoid misunderstanding, we have reformulated the text where subsaturation is first mentioned ("subsaturated conditions"). Then we have added a bracketed comment where we state that we use the convenient expression "subsaturated air" although we know that it is the water vapour that is subsaturated. Unfortunately, the opinion that the air has some kind of "water holding capacity" is quite popular, although it is entirely wrong.*

*L24f: Three very short sentences in a row. Maybe connect the last two sentences into one?*
*REPLY: Yes, no problem.*

*L45: „. . . it is not yet known which pathway dominates." Why should one process dominate?*
*REPLY: We think, in a given situation it can be known, which pathway dominates (and which time scale is the shorter one, see Bier et al., 2017), but if one asks this question in a general sense, that is, which pathway dominates globally and in a climatological sense, nobody knows the answer. This is meant here, and it might help to add the "globally and climatologically" to the sentence to be clearer. The text has been changed accordingly.*

*L62: how robust are the results on the ISSR definition of RHi>93%?*
*REPLY: Unfortunately, we cannot test that directly, since the 93% threshold is fixed in the output system that produces the WAWFOR-Klima data from the ICON output. Since we use then a binary field, which is obviously a non-linear function involving this threshold, there is no possibility to test the sensitivity. It is obvious, however, that the ISSRs shrink with increasing the threshold and vice versa. With shrinking ISSRs, lifetimes would diminish as well. However, we have some indications that the sensitivity around 93% is not strong. The second author made a corresponding test with ICON data for a conference at the Royal Aeronautical Society in 2021 ( Greener by Design conference "Mitigating the climate impact of non-CO2 – Aviation's low-hanging fruit" Gierens, 2021) For the presentation, he has tested, how the forecast of ice supersaturation and contrail persistence changes with the choice of the supersaturation threshold. He used 100, 99, 95, 90, and 80%. This is illustrated*

*in Fig. 1. It is evident, that there is a big difference when the threshold changes from 100 to 99% (panel b). However, the differences are moderate for tresholds below 99%. Even the difference between 90% and an 80% threshold are moderate (panel a). Thus we can assume that a choice of 93% is justified and the sensitivity of the results is moderate. This is now mentioned in the text.*

[Figure]

**Figure 1.** a) ISSRs at 250 hPa, valid at 22 Febr. 2021, 0700 UTC, for ISSR thresholds as indicated on the right. b) ISSRs at 300 hPa for the same date, for thresholds 100 and 99%.

*We can offer a further plausibility argument. In our previous paper (Hofer and Gierens, 2025) which uses the same threshold, we found a mean difference between the speed of an ISSR and the wind at the same location of about $(-5.3 \pm 11.8)\,m\,s^{-1}$, which corresponds to about $18 \pm 44\,km\,h^{-1}$. The mean pathlength of an aircraft within an ISSR is about 150 km (Gierens and Spichtinger, 2000), which implies that a contrail air parcel is on average 75 km away from the edge of an ISSR. Now, if we divide this distance by the mentioned speed difference, the result ranges from $1.2$ to $4.2$ hours. This is similar to the results derived in the paper. Please note, that the pathlength distribution is skewed and that much larger pathlengths than 150 km occur which lead to larger residence times, or synoptic time scales.*

*L65f: What is the vertical resolution of the underlying model?*
*REPLY: The vertical resolution of the WAWFOR data is quite appropriate for aviation, one flight level, i.e. 100 feet or about 60 m. There are 57 Flight levels (FL, in hecto-feet), from FL 50 to FL 600, and FL 675. These levels are interpolated from the high-resolution ICON output which has 120 vertical terrain following levels, which range from 75000 m (2.1 Pa) down to 20 m (101085 Pa) (Reinert et al., 2024). This information is added to the paper.*

*Eq (6): Parenthesis in equation is missing.*
*REPLY: Sorry, but the equation is correct.*

*Also, here you refer to ln, later in Fig. 5,8,11 you refer to log? Do you mean the same? Please clarify.*
REPLY: *Generally we use the natural logarithm in this paper and we write "ln" for it. But in software packages it is usually called as "log", and this led to the confusion. We will use "ln" throughout the paper now, that is we change the figure labels accordingly.*

*Sec. 3.2.1: I find it misleading to denote the time with a capital „T" and would suggest a small „t"*
REPLY: *If this would just be a running time (as a co-ordinate) we would have used a $t$. But our $T$ is a duration, the time span that an air parcel is within an ice supersaturated region. This time is thus not like a co-ordinate, it is for each air-parcel a single value. Therefore we prefer to stay with $T$. But we will write "time span" or "duration" to be clearer.*

*Sec. 3.2.3: Regarding Z\*. It would help the reader if the brief description of Z\* is provided already here and not in Sec. 4.3*
REPLY: *Yes, this is a good idea and a short description, why and how we normalise the geopotential is added at the beginning of sect. 3.2.3, where $Z^*$ is mentioned first.*

*L130: what is „high altitudes"?*
REPLY: *Yes, that's vague. We will instead write "upper troposphere".*

*L165: what is meant with „exclusively"?*
REPLY: *Yes, this is perhaps not a good expression. We will write instead that high normalised geopotential is present in the entire region (or similar).*

*Fig. 10: labeling missing ((a) to (f))*
REPLY: *Will be added.*

*Sec 4.1 feels out of place here. The discussion about the difference between lifetime and time-scales would be better placed in the introduction. The discussion of trajectory calculation, on the other hand, would also be useful in the methods section. Also: why are methods and results combined in one section?*
REPLY: *As the 2nd reviewer made a similar comment, we decided to delete Sect. 4.1 and to distribute part of its contents to a) the introduction (lifetime vs time-scale) and to Sect. 3.2.2 (potential errors in trajectory calculations). We agree as well with the 2nd part of the comment and now put the case studies in a section of its own (new section 4). That is the Discussion is now Sect. 5.*

*L 284f: what does „mainly" and „rarely" quantitative mean?*

REPLY: *We can of course quantify this. We could count the grid boxes with ISS at $Z^*$ larger or smaller than 0.98. In case studies 2 and 3, ISS does not occur at $Z^* < 0.98$. For these cases, "rarely" means zero. In case 1 it does occur at very few grid points where $Z^* < 0.98$, perhaps 3. This numbers are not representative as they refer to the selected situation. There is no gain in knowledge if such numbers are added to the text. Nevertheless, we have rephrased the text into a more quantitative form.*

*L 286: Will the prediction be easier if the normalised geopotential is calculated first in order to narrow down the calculation of the ISSR? This introduces a further work step.*
REPLY: *We think the further work step is almost negligible, since the normalised geopotential is merely a simple function of the geopotential that is output by any NWP model anyway. However, if air route planners can concentrate their efforts to avoid ice supersaturation to regions with large normalised geopotential, perhaps some unnecessary work can be saved there, more that needed to perform the simple functional transformation from $Z$ to $Z^*$. We have added a sentence to describe this at the end of this section.*

*Generally, please refrain from using statements such as 'we think' or 'eplains probably' as this leaves a lot of room for speculation.*

*REPLY: Yes, indeed, we agree, although even as scientists we are sometimes faced with issues that are not (yet) completely understood or quantified (or quantifiable). Sometimes it is necessary to formulate hypotheses (a more noble word for speculation), which then need to be tested. Anyway, we'll try our best to avoid such phrases.*

**Review 2:**

*The objective of this study is to identify the typical synoptic and microphysical time-scales that constrain contrail lifetime and inform the targeted use of alternative fuels. The manuscript uses hourly aviation weather forecast data based on ICON simulations from the German Weather Service. The model output of temperature and relative humidity is used in combination with the Schmidt-Appleman-criterion to estimate the potential for contrail formation, the persistence of contrails, and their tempo-spatial evolution over a 24 hour period. Information about the wind speed and wind direction is used to infer the advection of the detected ice supersaturated regions (ISSRs) and to calculate parcel trajectories.*
*The analysis is based on a total of three days in April and Mai 2024.*
*Based on this analysis, the authors find that in most of the analyzed cases, the synoptic and microphsyical time-scale are of similar length and argue that they are likely to be equally important. The authors further state that an efficient application of alternative fuels requires the microphysical time-scale to be smaller than the synoptic times-scale.*

*I do have some major comments, which mostly address the structure of Section 4.1, which could be clearer and in a better order. In general, a more concise writing, for example by avoiding jargon, can aid the understanding of the entire paper.*

*Besides the major comments there are a number of minor comments, mostly related to style, which does not follow the ACP submission guidelines. These could easily be avoided. Some of the comments are of a suggestive nature and could be considered by the authors. Other comments ask for more precise wording.*

*After the authors have adequately addressed the major and minor comments below, the manuscript may be considered for publication in ACP.*

**Major comments:**

*Section 4.1 is, in my opinion, unnecessary long. The authors discuss about potential definitions. This section could be much shorter and to the point by defining their time-scales. But this is already done in Section 3.1. Why not include Section 4.1 in Section 3 and define your time-scales and lifetimes there?*

*REPLY: As the 1st reviewer made a similar comment, we decided to delete Sect. 4.1 and to distribute part of its contents to a) the introduction (lifetime vs time-scale) and to Sect. 3.2.2 (potential errors in trajectory calculations).*

*L216: Regarding the comparison between observations and models. Isn't this a more fundamental question of several components? In both, the measurements and in the simulations, you have to define a cloud by some kind of threshold (see also minor comments L178ff). What is detected as a cloud depends strongly on the definition of a cloud, the threshold, and how and by what means the cloud is detected.*

*REPLY: Sorry, but we do not understand this question. In the current paper we do not use observations. The work by Dietz (2012) used a model as well. Neither we understand the comment on clouds, since we do not consider clouds in this paper, neither observed nor modelled ones. Otherwise, yes, the "boundary" of a cloud is fuzzy and what we label a cloud depends on what is possible in observations (some Lidar instruments can detect "clouds" with an optical thickness of $10^{-4}$) and on threshold settings in models. Nothing of this is a topic of our paper, so we do not understand the comment.*

*Second, the authors identify ISSRs, i.e., regions that allow the formation of persistent contrails. Contrarily, observing contrails from satellite, identifies and tracks contrails that actually formed. It would be good if the authors could comment on this.*

*REPLY: Yes, we agree. There are two essential differences.*

*1) In our method we isolate the synoptic effects, since the NWP model does not represent contrails. That is, we have ISSRs and air moving in and through them. In contrast, in a real contrail, all processes work simultaneously, which makes the distinction of synoptic and sedimentation time-scales impossible. We think, this is already discussed in the paper in the second paragraph of Section 4.2 (now 5.2).*

*2) In the model it is possible to follow a process up to the very end simply by looking at the output values. In contrast, it is difficult in nature to follow a contrail until complete dissolution. This holds for all methods, for satellite imagery, in-situ measurements with research aircraft, and ground observations. Finally, in a model one has the whole ISSR at hand, that is, one can determine its area, volume, ice and water vapour content, etc. Likewise, if a cloud or contrail is modelled, its properties are in principle completely knowable. This is not possible in nature, neither for ISSRs not for clouds and contrails. We added a few sentences at the endof the former Sect. 4.2 (now 5.2) and hope that it is what the reviewer suggested to think about.*

*Third, what is the effect of your threshold of RHi>93% for ice-supersaturation and the size of the ISSRs? To answer this question, the authors could vary the threshold for ice-supersaturation and check the robustness of the results.*

*REPLY: Review 1 had the same question and the answer is provided there.*

*In the second last section of the summary, the authors state that for targeted use of alternative fuels, trajectory calculations are required. If I am not mistaken, this would be required for several flight levels and over larger areas, where aircraft are flying and potentially interact with ISSRs. This can be computationally challenging. Perhaps the authors could provide a few comments on the following points:*

- *My interpretation of this approach is that it is a near real-time avoidance strategy that may be limited by airspace restrictions and air traffic control, as flight plans and routes are well defined days to months in advance. This may change in the future, but is difficult to apply today.*
  *REPLY: No, we think, although this is a real time strategy, it should not interfere with annual flight-plans and so on. If the pilot or the dispatcher or the flight-planner would know in advance and with some certainty that the flight will cross an ISSR and that it cannot be avoided without much costs or much additional emissions, the aircraft could be fuelled with alternative fuels instead of kerosene. As soon as sufficient amounts of alternative fuels would be offered at airports, this should be possible and the decision could be made shortly before the aircraft tanks are filled.*

- *Trajectory calculations are time consuming, requiring the calculation to be started well in advance of the flight day in order to adjust the flight path (if at all possible on short notice).*
  *REPLY: In fact, there are currently already services for contrail avoidance in place or under development. At least some of them use trajectory calculations to compute the advection of a contrail with the wind and the sedimentation of the ice crystals. Although we do not know how reliable these simulations are, they are obviously quick enough that flight plans can be based on them.*

- *The approach relies on the accuracy of the underlying weather model to predict the exact location of the ISSRs several days in advance, since it must include the time for the trajectory calculation plus the time to adjust the flight. REPLY: This is true and this is where we currently see the narrowest bottleneck for contrail avoidance and for mitigation of there impact using alternative fuels.*

*The authors could partially answer this by providing the times they needed to perform the calculations.*
*REPLY: See above. The second point has been added to the conclusion section.*

**Minor comments:**

*L20: " in air with sub-saturated water vapor" Please reconsider this phrase. Water vapor cannot be subsaturated by itself.*
220 *You are probably referring to the air, which can be subsaturated.*

*REPLY: Here you undergo a very popular error. Generally in thermodynamics the notion "saturation" refers to the vapour of a substance above its condensate, here water vapour above water or ice. If you consider any derivation of the Clausius-Clapeyron equation (which gives the change of saturation or equilibrium vapour pressure with temperature) you can see that*
225 *it refers to a single substance. Saturation simply means that there is flux-equilibrium between the H2O molecules in the vapour phase and an adjacent condensed phase. Thus, considering water vapour in the atmosphere, it is the vapour that can be saturated, subsaturated or supersaturated. Our sentence was written in this peculiar way with exactly the intention to be very clear on this point. Nevertheless, as the sentence was a bit ugly, we rephrased it using the expression "subsaturated conditions". Then we added in brackets a sentence that explains that, for convenience, we also use the misleading expression "subsaturated*
230 *air", although we know, that it is incorrect. Only the water vapour itself can be (sub/super)-saturated. The unfortunately common imagination of air having a "water holding capacity" is entirely wrong.*

*L27: Please clarify what you mean by "All of these effects should become stronger with increasing contrail lifetime". If one considers constant illumination conditions but a thinning of the contrail with time, as you show in the manuscript, then the radiative effect would decrease even though the contrail can be considered persistent. By rephrasing this sentence, you might*
235 *better explain your actual intent.*
*REPLY: It seems that we need to be clearer. Your example with constant illumination illustrates the misunderstanding that is possible here. You are right, that the radiative effect decreases over time, but its integral increases. So we change "should get stronger" into "accumulate".*

*L36: "dominate" instead of "dominates"?*
240 *REPLY: corrected*

*L38. "Fewer" instead of "less"?*
*REPLY: ok.*

*L56: "data inform". Data is singular, so I think it should be "data informs". You might consider rephrasing it, since data itself cannot inform. You could write something like: "based on the data, ...."*
245 *REPLY: Yes, we agree that data is singular and that they do not actively inform anybody. We will rephrase that sentence.*

*Fig1 and subsection: Please specify the unit of r0 on the y-axis. Furthermore, if the isolines represent the time-scale, why did the authors choose an example that is "not included" in the figure. It would be more intuitive and easier for the reader to choose an example with a time-scale of 4.5h or 6h, since these lines are included. Conversely, you could add the 5h isoline. Does the word "deep" refer to the geometric thickness of the cloud?*
250 *REPLY: Please note that $r_0$ is a pure number (see line 85). We agree to select isolines that better illustrate the example. Finally, "deep" will be replaced by an expression involving "geometric thickness" or "vertical extension".*

*Eq.6 Please check that the double "ln" is correct.*
*REPLY: Yes, it is correct. If you look at the formula of the Weibull-type survival function (eq. 4), it is clear that one needs to apply the logarithm twice to solve for the exponent $k$.*

255 *L135-136: Instead of "transported downwards to the ground" the term "subsidence" could be used, which is shorter and more common in this context and refers to large-scale sinking air masses.*
*REPLY: Yes, we agree, and we also feel that the word "ground" is not appropriate here, where we simply mean downwards.*

*Fig4. You might consider adding (a) and (b) to the panels to make them more easily referenced in the caption and text.*
*REPLY: Yes, agreed.*

260 *L146: Instead of qualitative measures, it would be better to quantify and provide absolute numbers and/or percentages of how many air parcels remained ice-supersaturated.*
*REPLY: We follow your suggestion. We could count the green-marked grid points, or use the survival function and its parameters. The latter method gives $0.75$, that is after one hour 75% of the initially supersaturated air parcels are still supersaturated. After 24 hours more than 99% of the air parcels are subsaturated.*

265 *Fig5.: Equation 5 uses "ln", which has a base of e. However, the y-axis in Fig.5 uses "log", which usually has a base of 10. Please check for consistency.*
*REPLY: Good point!*

*L154: Here and elsewhere in the text, e.g., L67: Please check the date format here, in the caption of Figure6, and compare with the ACP submission guidelines. Copied from the ACP submission guidelines: "Date and time: 25 July 2007 (dd month*
270 *yyyy), 15:17:02 (hh:mm:ss). Often it is necessary to specify the time if referring to local time or universal time coordinated. This can be done by adding "LT" or "UTC", respectively. If needed when referring to years, CE (common era) and BCE (before the common era) should be used instead of AD and BC since CE and BCE are more appropriate in interfaith dialogue and science."*
*REPLY: Yes, we will adhere to the style guidelines.*

275 *L159-161: You might consider quantifying "many" and "almost all".*
*REPLY: Yes, this is similar to the comment above. We provide relative values for all three cases.*

*L161: Please quantify "excellently". For example, by providing an r-squared for the fit.*
*REPLY: Of course, one could calculate $r^2$, but to our view, in fig. 8 one can clearly see that the fit is quite good, and we doubt that we learn essentially more by providing numbers. As a reaction to this comment we delete the word "excellently".*

280 *L178-179: The phrase "...can no longer be seen..." is imprecise. "Visibility" of contrails and water vapor is strongly dependent on wavelength. Thus, the authors would need to provide a wavelength range in which the observation takes place. Please see Driver et al. 2025, where they state that there is no clear threshold (https://amt.copernicus.org/articles/18/1115/2025/). It might be better to provide a threshold to describe the presence of contrails, such as, ice mass.*
*REPLY: As the old Sect. 4.1 has been deleted in the revised version, the comment does no longer apply.*

285 *L179: "In principle, such a period can precisely be determined,..." I doubt that this is an easy task. As mentioned before, it is almost a philosophical question what counts as a cloud or water vapor. Clouds that are optically thin at solar wavelengths can still be very effective at absorbing radiation in the thermal infrared. To make such a judgment, a threshold must be used. Since, to my knowledge, there is no consensus on such a threshold, the authors may define their own threshold based on reasonable judgment.*
290 *REPLY: As the old Sect. 4.1 has been deleted in the revised version, the comment does no longer apply.*

*L181: "... (another vague word) ..." What is the authors intention of this insert? The first paragraph of this section could be omitted, or at least shortened, by providing a clear definition of the time-scale as understood and defined by the authors.*
*REPLY: As the old Sect. 4.1 has been deleted in the revised version, the comment does no longer apply.*

*L198: "One might believe..." and ".. but we do not think so" are expressions that should be avoided in scientific publica-*
295 *tions. Hypotheses and conclusions are based on reasoning and facts. Please consider rephrasing this sentence.*
*REPLY: We agree. But this peculiar expressions are no longer present, as Sect. 4.1 has been deleted.*

*L216-219: Please check the grammar in this sentence*
*REPLY: We rephrased the text.*

*L219-222: Please consider rephrasing this sentence. What you probably mean: There are two types of trajectory calcula-*
300 *tions: 1) following the air parcel in normal mode, and, as I understand it, 2) following the air parcel but with precipitation /*
*sedimentation and condensation switched off. This is mentioned later in L231, but would fit here.*
*REPLY: Yes, we agree and thank for the suggestion.*

*L219-222: The term of "virtual" humidity may be an unfortunate choice, as it may be confused or associated with other*
*metrics such as virtual temperature. You might consider calling it along the lines of "idealized simulations". However, this is*
305 *just a personal impression and I would not insist on a change.*
*REPLY: We agree, but we cannot avoid the word "virtual" here since it was used by the author of the thesis we refer to. We*
*replace "calculation" with "idealized simulations" in the text to make it clearer.*

*L225 and 226, RHi should be "RHi" for consistency*
*REPLY: Yes, we agree.*

310 *L249ff: the word "height" is used, which is misleading. I assume you mean cirrus / cloud geometric thickness, right?*
*REPLY: Yes, you are right. We corrected "height" into "thickness".*

*L312: The authors please better explain what they mean by "real world". A better term might be "in nature"?*
*REPLY: Yes, we agree and thank for the suggestion.*

*L315: "soor-poor-regime" is mentioned here for the first time. The authors could mention this earlier when introducing the*
315 *targeted use of alternative fuels.*
*REPLY: Yes, we agree. This is a special notion and should be explained and the reader should not expected to read another*
*paper.*

**For the entire text:**

- *If I am not mistaken, the first appearance of the figures in the text follows: 3, 4, 5, 2, 6, 10, 8, 11, 1, and 12. Consequently,*
320 *the inclusion of the figures should be reordered. Figures 9 and 11 are not referenced anywhere in the text. Either refer to*
*them in the text or, if they are not needed, remove Figures 9 and 11.*
*REPLY: indeed the figure numbers are a mess. We checked this and corrected it accordingly.*

- *The authors are inconsistent in their use of "Section", "section", and "Sec." Copied from the ACP submission guidelines:*
*"The abbreviation "Sect." should be used when it appears in the running text and should be followed by a number unless*
325 *it is at the beginning of a sentence."*
*REPLY: This will be corrected.*

- *"Figure" is written at the beginning of a sentence and abbreviated within a running sentence. Copied from ACP submis-*
*sion guidelines: "The abbreviation "Fig." should be used when it appears in running text and should be followed by a*
*number unless it comes at the beginning of a sentence, e.g.: "The results are depicted in Fig. 5. Figure 9 reveals that...".*
330 *REPLY: This will be corrected.*

- *Please be consistent in your use of the serial comma if you choose to use it.*
*REPLY: This has been corrected.*

335     – *Please be consistent in your use of units. Sometimes "hours" is used and sometimes "hrs". Check the ACP guidelines.*
      *REPLY: This will be corrected.*

    – *Variables are usually in italics, while subscripts are in normal font. For example: instead of $RH_i$ write $RH_\mathrm{i}$*
      *REPLY: This will be corrected.*

340

**References**

Bier, A., Burkhardt, U., and Bock, L.: Synoptic control of contrail cirrus lifecycles and their modification due to reduced soot number emissions, J. Geophys. Res., 122, 11 584–11 603, https://doi.org/10.1002/2017JD027011, 2017.

Dietz, S.: Untersuchung charakteristischer Lebenszyklen von eisübersättigten Regionen in der oberen Troposphäre, Master's thesis, Universität Innsbruck, 2012.

Gierens, K. and Spichtinger, P.: On the size distribution of ice–supersaturated regions in the upper troposphere and lowermost stratosphere, Ann. Geophys., 18, 499–504, 2000.

Gierens, K. M.: Contrail Statistics, Big Hits and Predictability, in: RAeS Conference: Mitigating the climate impact of non-CO2 – Aviation's low-hanging fruit, https://elib.dlr.de/141532/, 2021.

Hofer, S. and Gierens, K.: Kinematic properties of regions that can involve persistent contrails, EGUsphere, 2024, 1–23, https://doi.org/10.5194/egusphere-2024-3520, 2025.

Reinert, D., Prill, F., Frank, H., Denhard, M., Baldauf, M., Schraff, C., Gebhardt, C., Marsigli, C., Förstner, J., Zängl, G., and Schlemmer, L.: DWD Database Reference for the Global and Regional ICON and ICON-EPS Forecasting System, Version 2.3.1, Tech. rep., Deutscher Wetterdienst (DWD), 2024.

Wilhelm, L., Gierens, K., and Rohs, S.: Meteorological conditions that promote persistent contrails, Appl. Sci., 12, 4450, https://doi.org/10.3390/app12094450, 2022.